# Acquiring a Pet Dog: A Review of Factors Affecting the Decision-Making of Prospective Dog Owners

**DOI:** 10.3390/ani9040124

**Published:** 2019-03-28

**Authors:** Katrina E. Holland

**Affiliations:** Dogs Trust, 17 Wakley St, London EC1V 7RQ, UK; katrina.holland@dogstrust.org.uk

**Keywords:** dogs, puppies, pets, puppy acquisition, dog acquisition, human behavior

## Abstract

**Simple Summary:**

Each year, many people around the world get a pet dog. With so many different types and breeds of dogs available, and a variety of sources from which to obtain a dog, the process of getting a dog can be complex. The decisions involved in this process are likely influenced by a variety of human- and dog-related factors and this review explores the factors that appear to be the most important.

**Abstract:**

Given the prevalence of pet dogs in households throughout the world, decisions regarding dog acquisition affect many people each year. Across the stages of dog acquisition there is potential for practices that may promote or compromise canine welfare. For instance, prospective owners may not fully understand the time, energy and financial commitment entailed in their decision to acquire a dog. Thus, it is pressing that stakeholders, including those working in the canine welfare sector, refine their ability to identify and respond to trends in the behavior of potential dog owners. The motivations, attitudes and behaviors of current and prospective dog owners is a small but growing area of interdisciplinary study. Yet, no synthesis of the evidence exists. To address this gap, this critical review collates data and insights from studies published by academic researchers and animal welfare charities. The most widely reported factors associated with acquisition behavior include: the dog’s physical appearance, behavior and health; social influences, such as trends in the popularity of certain breeds; demographic and socioeconomic factors; and the owner’s previous ownership experience. Overall, the research discussed in this paper highlights that complex interactions likely underpin the various factors that might influence prospective owners’ motivators and behaviors.

## 1. Introduction

In many parts of the world, pet dogs are highly prevalent. Dog-ownership is reported in 49% of US households [1], 39% of Australian households [2] and 26% of UK households [3]. The acquisition of a dog is thus an experience that many people will undergo, potentially multiple times in their lifetime. Broadly speaking, acquisition refers to the process owners go through when obtaining a dog, from early decision-making through to the act of purchase or adoption. Hence, the decision whether to get a dog is only one step on the road to acquiring an animal. Other factors that prospective owners might consider include dog breed, type (e.g., fighting, herding, hound, lap, ratters, sporting, working [4]) and physical or behavioral characteristics pertaining to the individual animal chosen. Furthermore, the prospective owner must also decide from which of the various sources (e.g., breeder, third-party seller or rescue center) to acquire their dog. While this paper focuses on the various stages of the process of dog acquisition, it is important to note that not all owners experience an active decision-making process prior to acquiring their dog: for example, some may instead act on impulse based on opportunity. Under these circumstances, the owner has not made decisions regarding breed/type or from where to acquire the dog.

It has been suggested that the practices associated with the breeding and selling of dogs are tied, in various ways, to the health and welfare of these animals. For instance, a range of health conditions and hereditary diseases have been identified in dogs who have been selectively bred for exaggerated physical features, including excessive facial skin folds, flat faces and protruding eyes [5]. With regards to the sources of dogs, concerns have been raised about the motivations of some involved in the puppy trade. While many breeders and sellers undoubtedly care for the welfare of the animals in their care, the UK government (Defra) suggests that there are others for whom the dogs’ welfare is considered secondary to optimizing profitability [6]. For example, concerns have been expressed that the selling of dogs by some third-party sellers (sellers who have not bred the dog themselves) might lead to the early separation of puppies from their mothers and a lack of socialisation and habituation to people and other animals.

Furthermore, each year, many dogs are relinquished globally [7,8]. Owners who give up their dogs do so for various reasons and sometimes relinquishment is influenced by external factors, such as an inability to find rental accommodation that will accept a pet [9]. However, owner expectations—including a lack of appreciation of the time, effort and costs involved in dog ownership—are also a frequent reason for the surrender of dogs to a rescue shelter [10,11,12,13]. Thus, some cases of relinquishment might be avoidable by preventive measures (e.g., education, campaigns and policy) targeted at promoting responsible acquisition behavior. Greater understanding of how people are making decisions about dog acquisition is needed to develop such strategies. 

This report summarizes available evidence regarding owners’ motivations and behaviors during the process of dog acquisition. Where data are available, the possible role of demographic and socioeconomic factors in the process is also considered. In presenting the current research on dog acquisition behavior, the limitations and knowledge gaps of these studies are highlighted. In addition, several topics for further study are identified that would support the design of appropriate interventions and inform best practice for a variety of stakeholders including policy-makers, breeders, rehoming organisations and advisory bodies regulating the trade of pets.

The paper is divided into sections summarizing the evidence regarding four key areas in the process of dog acquisition: (1) the decision to acquire a dog; (2) the decision about what type of dog to acquire; (3) the decision about which individual dog to acquire; and (4) the decision regarding which source to acquire a dog from. While these stages are discussed here sequentially, for ease of discussing the research, in practice it is likely that the sequence of the stages varies widely between owners, both in terms of the order and pace at which the process is experienced and, as noted above, in some instances one or more stages may not occur at all.

## 2. Choosing to Acquire A Dog

### 2.1. Demographic and Socioeconomic Factors

#### 2.1.1. Household Structure

Several studies have indicated that the number of people in a household is a key variable in predicting dog ownership [14,15]. Westgarth et al. [16] identified households with five or more occupants as being more likely to own a dog than those with fewer members. This finding has been supported by a cross-sectional study of pet owning households across the UK, in which it was reported that the likelihood of dog ownership increased as the number of people within a household increased [17]. Possibly in relation to this trend, studies in the UK [16], USA [18,19], Ireland [20], and Canada [21] have reported that dogs are more likely to be owned in households containing children. Some studies have indicated that the age of the children might be important in indicating the likelihood of dog ownership. Westgarth et al. [16] concluded that households without children aged five years or younger were more likely to own a dog than those with children aged five years and younger. Similarly, Murray et al. [17] reported that families with children aged 10 years or younger were almost half as likely as households without children in this age group to own a dog. 

#### 2.1.2. Accommodation Type

Westgarth et al. [22] determined that people living in detached houses were more likely to own a dog than those living in other types of dwelling. Regarding the ownership of pets in general, rather than dogs specifically, findings from a Swedish study demonstrated that pet owners were more likely to live in a house than non-pet-owners, who more commonly lived in an apartment [23]. However, in a study of households in a single English county, no significant association was found between accommodation type and dog ownership [16]. The authors posited various suggestions to account for this finding that contradicts other research, including the theory that the variation might reflect genuine differences in the different study samples, and/or inadequate power of the study [16] to detect a difference in a small and relatively homogenous study area. In addition to housing type, the status of home ownership has also been observed to impact the probability of dog ownership in the United States, with significantly greater rates of ownership found amongst home owners compared to renters [14,15].

It is worth noting that these studies do not consider dog ownership amongst homeless populations. Given that, in the USA, an estimated 6–24% of homeless persons own pets [24,25] future research would benefit from a focus on the acquisition process as experienced by people without a fixed address.

#### 2.1.3. Socioeconomic Status

The relationship between dog ownership and socioeconomic status (SES) has been investigated in several studies. While there is no universally accepted definition of SES, broadly speaking it is an indication of an individual or family’s social and economic position in relation to others. Unlike variables such as height or weight, SES cannot be directly measured. As a result of this difficulty, in the fields of social and behavioral research, approximations of SES are typically based on proxy variables of factors including income, educational achievement and occupation.

One of the most widely used indicators of SES—household income or family affluence (typically based on material markers, such as the ownership of a car)—has been variably reported in relation to the probability of owning a dog. While some American studies focusing on ownership of pets in general have identified a trend for higher household income predicting pet ownership [19,26,27,28,29], Marsa-Sambola et al. [30] reported that adolescents from high affluence families were less likely to own a dog than those from low-affluence families. In contrast, Murray et al. [17] found no significant association between household income and dog ownership. 

Another variable that has been considered as a proxy for SES is the owner’s level of education. Based on multivariable analysis of data from households across the UK (*n* = 2980), Murray et al. [17] reported that the likelihood of dog ownership decreased as the highest level of academic qualification obtained by a household member increased. Similarly, a meta-analysis of cohort-studies found that a higher educational level significantly reduced the odds of dog ownership in European families [31]. Highlighting the complexity of the relationship between dog ownership and owner education, using data from a UK birth cohort, Westgarth et al. [22] modelled the interaction of current dog ownership, parental education and pet ownership during the parents’ childhood. Specifically, they reported an interaction between the mother owning pets during childhood and paternal education; in cases where the mother had never had pets as a child, as the level of paternal education increased the probability of dog ownership decreased. However, when the mother had owned pets in childhood, this effect was not observed. 

A further socioeconomic measure, the effect of owner occupation, has been variably reported with regard to the likelihood of dog ownership. In multivariable analyses in the UK [16] and Canada [21] it was shown to be non-significant. However, in Sweden, a lower measure of socioeconomic status based on a classification of occupations has been associated with a greater prevalence of dog ownership [32].

The differences between the findings of these studies might be, at least in part, explained by the different measures used to assess income, education and occupation. The variability also suggests that the role of factors such as income is likely to be inter-dependent with other influences. Future research should look to develop insights into the interaction of the multiple variables thought to be associated with dog ownership.

#### 2.1.4. Ethnic Variation

Studies have reported a greater prevalence of dog ownership among individuals of white ethnicity in samples of 9- to 10-year-old children in Liverpool (England) [33] and 11- to 15-year-old adolescents in Great Britain [30] than their peers from other ethnicities. Ethnicity, as a factor influencing dog ownership, has also been explored among young adults (with a median age of 25.90 years) in a sample of American university students [34]. In this study, white students owned significantly more pets than their African-American peers. Religious, historical and cultural traditions and beliefs likely influence attitudes towards dogs [33] and this area would benefit from future study to evaluate which factors influence dog ownership within different ethnic groups. 

### 2.2. Prior Dog Ownership

Several studies have demonstrated a correlation between a person’s contact with pets during childhood and their desires and behaviors towards pet ownership in adulthood. For instance, in a cohort study that analysed pet ownership data collected from mothers of 14,663 children, Westgarth et al. [22] reported a positive association between dog ownership and prior pet ownership: whether the mother had pets during her childhood was a strong predictor of pet ownership. Similarly, in a small-scale survey of UK university students (*n* = 385), a significant positive association was observed between a person’s contact with pets during childhood and their reported likelihood of keeping pets as an adult [35]. Given that this study’s sample was comprised of students, a limitation is its reliance on participants’ speculation about their pet-keeping aspirations rather than data reporting their actual behavior. However, when considered alongside the results of several other studies [36,37,38], an observation that pet ownership in general is more common among people with historical pet ownership becomes apparent. A more recent study that looked at dog owners’ prior experience with dogs observed that this experience was often influential in their subsequent practices surrounding dog acquisition [39]. This study, conducted with 255 dog owners in Washington, USA, found that 43% of owners reported that their decision to adopt a dog was mostly influenced by their prior experience with dogs. Regarding decision-making around dog breed choice, the owner’s previous dog experience was found to be of even greater importance, with 57% of respondents reporting that this factor influenced their decision of breed most. 

The research discussed in this section suggests that the decision to acquire a dog is frequently influenced by previous dog ownership [22,36,37,38,39]. Other factors that are suggested to be associated with a greater likelihood of dog ownership include a larger number of household occupants [14,15,16,17], the presence of children in the household [16,18,19,20,21] and white ethnicity [30,33,34]. Several socioeconomic and demographic factors, such as household income [17,19,26,27,28,29], occupation [16,21,32] and accommodation type/ownership status [14,15,16,22,23] have been variably reported with regards to their impact on dog ownership. It is unlikely that the relationship between these individual variables and the decision to acquire a dog is linear, which may help to explain the variation across the studies. Future exploration of the process of dog acquisition would benefit from greater attention to the interactions among these factors.

## 3. Choosing Which Type of Dog to Acquire

### 3.1. Dog-Related Characteristics

#### 3.1.1. Canine Appearance

When deciding which type or breed of dog to acquire, one factor that might be influential is the dog’s physical appearance. The importance of the dog’s appearance has been reported to manifest in various ways. For instance, several studies have reported that for some dog owners, the animal’s appearance may be considered more important than its health. In an American study, researchers found no correlation between behavior, health, or longevity with breed popularity [40]. On the contrary, the most popular breeds tended to have significant health problems, and possibly more behavioral issues. This finding prompted Packer, Murphy and Farnworth’s [41] (p. 199) assertion that ‘health considerations have been secondary in the decision to acquire dogs’. Among Finnish dog owners, physical appearance was found to be a more important factor in affecting their decision to acquire a specified breed when compared to the risk of serious breed-associated genetic diseases [42]. Similar findings were reported by Weiss, Miller, Mohan-Gibbons and Vela [43] who explored the motivations involved in dog acquisition among people (*n* = 1491) adopting dogs from animal shelters. In this study, a greater percentage of adopters indicated the dog’s physical appearance as important (75%) when compared to the proportion of respondents who rated the dog’s health as important (49%).

While the notion that many owners consider the dog’s physical appearance to be highly important in their decision-making is widely reported, when each physical trait is considered in isolation, the significance varies. For instance, the color of the dog’s coat was reported as “unimportant” for 65% of Australians asked to rate the importance of various characteristics pertaining to their perception of the “ideal dog” [44]. A similar finding was reported in a study conducted in Italy, in which 76% considered the dog’s coat color unimportant [45]. The same study found that the hair type and length were unimportant to 46% of participants. In Australia however, only 23% of participants considered hair type and length unimportant [44]. Of this sample, the most preferred hair type was short and straight (29%) Regarding size, medium-sized dogs (10–20 kg) were most popular, preferred by 40% of respondents. In Italy, the most preferred size was large (33%) followed by medium (28%). In another study, photographs of dogs with colored irises (notably human-like traits) were preferred by 73% of participants [46]. In this study, dogs displaying a distinct “smile” were also favored. The preference for such human-like attributes has been described as anthropomorphic selection—‘selection in favor of physical and behavioral traits that facilitate the attribution of human mental states to nonhumans’ [47]. More research is required to further understanding of the interaction of the appearance-related characteristics that are influential in the selection process. Some of these findings suggest the unimportance of various physical characteristics [44,45], diverging from Weiss et al.’s study in which appearance is reported as a determinant criterion for adopting a shelter dog [43]. This discrepancy might be explained by the notion that the thoughts people have about their ideal dog may clash with their behavior. For instance, while some people might prefer to own a large dog in an “ideal” world, their housing restrictions might limit them to owning a small one.

The importance of the dog’s physical appearance, relative to other factors, such as breed and health, may vary between owners of different breeds. This observation was indicated in a recent UK study that considered whether owners of brachycephalic (i.e., short-muzzled) breeds differed from those who own non-brachycephalic dogs, in relation to the motivational factors that attracted them to the breed type [41]. Appearance was identified as being of greater significance in influencing surveyed owners’ decisions to purchase their chosen breed for owners of brachycephalic breeds (specifically, French Bulldog, Pug and Bulldog breeds) when compared to non-brachycephalic breeds. Breed health, however, was of greater influence in the decision to acquire the chosen breed for owners of non-brachycephalic dogs. A Danish study also reported variation in acquisition-related motivations between owners of different breeds [48]. This study surveyed owners (*n* = 846) of four different dog breeds (Cavalier King Charles Spaniel, French Bulldog, Chihuahua and Cairn Terrier) about their motivational drivers in acquiring their choice of dog. The findings indicated that the importance of good health and physical appearance varied for owners of the different breeds. For example, owners of Cavalier King Charles Spaniels and French Bulldogs seemed to specifically value the distinctive appearance of the breed they chose, more than the owners of the other breeds. For owners of French Bulldogs and Chihuahuas, breeds that are prone to health problems owing to their extreme body conformation, the health of the breed was reported to be of less importance in pre-acquisition motivations when compared to the dog’s appearance. It was suggested by Sandoe et al. [48] that the lesser importance placed upon health compared to appearance among owners of some breeds with intrinsic health problems might not actually be a paradox at all. Instead, they suggest that the increased levels of caregiving behavior required to care for dogs with health and behavior problems might be another explanation for why owners of specific physically distinctive dog breeds report strong attachment ties to them. Thus, it is possible that health, as a trait, is potentially important in some owners’ acquisition motivations, though here it is poor health that is being favored. 

Evidence also suggests that owners of brachycephalic breeds might differ when compared to owners of non-brachycephalic dogs in their purchasing behaviors [41], with owners of brachycephalic dogs less likely to display recommended purchasing behaviors (i.e., those embedded within the Puppy Information Pack developed by the British Veterinary Association Animal Welfare Foundation and Royal Society for the Prevention of Cruelty to Animals [RSPCA] Puppy Contract) when purchasing their dog. For example, owners of brachycephalic dogs were less likely to see either parent of their puppy: 12% of brachycephalic owners saw neither parent, compared to 5% of non-brachycephalic owners. Those who owned brachycephalic dogs were also less likely to ask for any health records, suggesting that owners of these dogs are less motivated to buy a healthy individual within a breed. The authors suggest that this finding might further reduce the importance placed on health by brachycephalic breeders, by diminishing demand for healthy, tested dogs. Furthermore, while there were only a few reports of owners of brachycephalic dogs acquiring their dogs as an impulse purchase, owners of these dogs were more likely to have purchased their dog during the first and only visit to their breeder. They were also more likely to have found their dog through puppy-selling websites.

There are a variety of possible explanations that might account for the value placed on appearance in some people’s acquisition decisions. One possibility is that an attraction towards certain physical qualities of dogs might reflect a common human preference for infantile features (e.g., flat faces and large and wider-set eyes) in animals. According to the ethologist Lorenz, such attributes are understood to help make animals appear “cute” and appealing to humans, motivating their desire to care for the animal [49]. Empirical research, using pictures to evaluate the appeal of infantile facial features in dogs, supports Lorenz’s claim [46,50]. This bias towards infantile features might help explain the extreme aesthetic traits prevalent among dogs of certain breeds; especially those that are considered brachycephalic. In many cases, these features are the product of inbreeding, and the positive selection for extreme traits, or an absence of selection pressure against health issues [48]: a process that has been argued to have helped cause various confirmation-related chronic and severe health issues [5]. 

Despite the increased predisposition to health conditions and inherited diseases, some of these breeds are increasing in popularity. A chief example is the French Bulldog: a breed that was the second most popular dog breed registered in the UK in 2017 [51], having risen four places since 2013 [52]. This breed has undergone a similar rise in popularity in the USA too: in 2013 the breed was ranked 11th in the American Kennel Club’s (AKC) popularity rankings, but in 2018 it was reported fourth [53]. The attraction to an infantile aesthetic might help account for the apparent paradox of human behavior that is reflected in the popularity of dog breeds with extreme physical features and associated welfare problems. Indeed, the ability of puppies to eclipse rational human thought, due to the “cute response” is identified by the RSPCA [54] as one of the factors that creates the impulse to acquire a puppy. Regarding the increasing popularity of brachycephalic breeds such as the French Bulldog, Packer et al. found that owners who kept brachycephalic breeds tended to be younger, buying for the first time and without any prior ownership of dogs [41]. Further research is required to explain why this effect occurs, however the authors speculate that it might be associated with increased media influence among younger people. The role of social and media influences, such as celebrity endorsements, in decision-making have not been fully investigated, and future studies of owners of brachycephalic breed dogs in particular may provide insights into whether and how celebrity endorsement influences perceptions of brachycephalic dogs and their acquisition.

Another hypothesis that might partly explain variation in owner motivations surrounding the dog’s appearance and breed type chosen is associated with differences in owners’ personalities [55]. Based on interviews conducted with dog owners (*n* = 7), two distinct categories of motivation for pet ownership—intrinsic and extrinsic—were identified. People who are intrinsically motivated prefer to achieve goals that are more innately satisfying, supposedly valuing their pets for the sake of the individual animal itself. Meanwhile, those who are extrinsically motivated display behavior that earns external rewards and social acknowledgment (i.e., status). Thus, the authors propose that people who fall into the latter group are more likely to acquire fashionable dog breeds, commonly those of distinctive physical appearance that may influence social acknowledgment. The study authors also suggested that owners who are extrinsically motivated, or who acquire their dog as part of a personal identity project, tend to own “designer” dogs (e.g., Pugalier [Pug-King Charles Cavalier cross]) and purebreds (e.g., Pug). Such dogs are typically considered of “cute” or “toy” appearance. Meanwhile, owners with an intrinsic motivation towards dog ownership were more likely to own a non-purebred dog and were more concerned with their dog’s innate qualities than their appearance. The study’s findings must however be interpreted with some caution due to the exploratory nature of the methods used, and as they relate to a largely female and predominantly young (most were in their 20s) sample from one Western country.

In addition to owner personality, the findings of an American study suggest that the dog’s physical appearance varies in significance as an influential factor among owners of different demographic categories [56]. For instance, the study found that a higher percentage of men (68%) than women (61%) rated appearance as an important characteristic when acquiring a dog. 

#### 3.1.2. The Dog’s Age

Another characteristic that research has indicated is potentially influential in people’s decisions regarding the kind of dog they wish to acquire is the animal’s age. Based on adoption records (*n* = 1264) from two dog shelters in New York, length of stay in the rescue environment was observed to increase with the dog’s age [57]. Additionally, studies conducted in Australia and Italy, exploring the characteristics that constitute the general public’s perception of the ideal dog, found that there was a strong inclination towards acquiring a dog as a puppy [44,45]. However, while this could be associated with a preference for infantile-like features, it is not known whether the preference for younger dogs is directly linked to the puppy’s appearance or whether it is due to other factors, such as a perception that puppies will not have acquired bad habits, or an owner’s desire to experience the “puppy stage” of a dog’s life.

### 3.2. Trends in Breed Popularity

The influence of fashion on consumer choices is ubiquitous in society and research suggests that this social factor also affects people’s decisions regarding their choice of which dog breed to acquire [40,58]. For example, analysis of AKC registrations between the years 1946–2003, identified that the patterns of popularity of some dog breeds resembled other cultural variants such as baby names, as their popularity followed boom-bust patterns of growth and decline [58]. On average, the increase (boom) phase in the dog breeds that exhibited such patterns lasted around 14 years before a decline occurred. 

When fashion trends affect owners’ decisions regarding dog acquisition, it is possible that there might be implications for canine welfare. As Herzog suggests, fads in breed popularity may be a factor in the relinquishment of pet dogs, because social contagion may lead individuals to select a puppy that, when fully grown, is unsuited to the owner’s lifestyle [58]. Canine welfare is also at risk of being compromised by unscrupulous breeders who are capitalizing on consumer demand for such “fashionable” breeds. The financial incentives of producing these animals in high volume are clear: in the UK, French Bulldogs, can sell for up to £1570 per puppy, according to figures reported by the European Union (EU) Dog and Cat Alliance [59]. Some of the irresponsible breeding practices that might be engaged in by individuals hoping to profit from high consumer demand include excessive levels of inbreeding, bitches being bred too many times per year, and not socialising puppies [60]. An additional concerning practice that has been associated with the high demand for fashionable breeds and the limited supply of such puppies from reputable breeders is “puppy smuggling”, or the illegal importation of puppies. Several investigations have highlighted concerns about the trade in underage puppies and pregnant bitches being transported illegally from Central and Eastern Europe into Great Britain [61].

The media, including film releases, might intensify the development of fluctuations in breed popularity [62]. For instance, in the eight years following the 1985 re-release of 101 Dalmatians, the annual number of new AKC Dalmatian registrations increased from 8170 puppies to 42,816 [58]. This “dog movie star” effect has been found to last many years beyond a film’s release, with first day ticket sales significantly correlated with increases in breed popularity even 10 years later [62]. The impact of the media on breed popularity has also been identified in the UK, when the Kennel Club reported a 1,407% increase in online puppy searches for the White Swiss Shepherd Dog in the month following the launch of a television advertisement which featured this breed [63]. Hence, it is sometimes possible to identify specific underlying causes to trends in dog breed popularity. Subsequently, this knowledge could enable stakeholders to anticipate behavior changes and respond more appropriately.

However, not *all* media exposure has been found to correlate with an increase in breed popularity. For instance, Herzog and Elias [64] report that the Westminster Kennel Club Dog Show, despite being watched by millions of viewers on television each year, does not usually result in a substantial increase in new registrations of the winning breed. The variability in the reported effect of the media on acquisition behavior, might be an effect of possible differences in demographic factors of the respective audiences of popular dog-based films and the Westminster Kennel Club Dog Show.

### 3.3. Pre-Purchase Behavior

A small body of research has indicated that people’s decisions in relation to the acquisition of a dog may be influenced by various sources of information that are likely to vary in content and quality. A survey found that 40% of dog owners spent less than one week conducting research prior to purchasing [54]. Other research has indicated that around one fifth of prospective dog owners do not carry out any research at all before taking on a dog [65,66]. Kuhl [65] suggests that owners may consider themselves to have adequate experience not to require further research. However, these figures may also highlight the prevalence of impulse purchase behavior around pet dogs. Further research is required to reveal the factors that impact on the likelihood of pre-purchase research prior to acquisition.

Of those owners in Kuhl’s study who did carry out pre-acquisition research, the most popular sources of information were general internet searches (19%), books (16%), breeders (12%) and friends and family (11%). Many owners reported verbal communication (i.e., face to face conversations with vets or breeders) to be their preferred method of receiving information (56%) followed by emails (16%) and posts on websites, blogs and forums (16%). Data from the most recent People’s Dispensary for Sick Animals (PDSA) PAW (PDSA Animal Wellbeing) report (2018) [66] indicates the significant role of the internet in relation to pre-purchase behavior: 36% of the dog owners surveyed (*n* = 2080) reported that they used the internet to carry out research prior to choosing their dog. This figure has increased by 7% since 2015 [67]. Several topics for future study in this area can be identified. Firstly, a critical appraisal of the online sources generated by pre-purchase internet searches would be useful towards understanding the accuracy and readability of the literature most frequently consulted by prospective owners. Furthermore, there is scope for further study to explore how demographic and socioeconomic factors are associated with variations in pre-purchase behavior. While PDSA [68] data indicates that 25–34-year-olds are most likely to use the internet for pre-purchase advice regarding pets in general, more data regarding the interaction of multiple demographic variables would likely help stakeholders develop and target appropriate strategies in education and human behavior change.

The research that has been considered in this section highlights several factors that might inform owner’s decisions about the type of dog to acquire. Evidence suggests that, in addition to a preference for puppies over adult dogs [44,45,57], the physical appearance of the dog is important to many owners and may be a more significant factor than the dog’s health in shaping decision-making [42,43]. The importance of appearance has been observed to vary in relation to owners of different dog breeds, with this factor suggested to be particularly important for owners of brachycephalic and/or fashionable breeds, such as French Bulldogs [41,48]. Attempts to explain the importance of appearance include theories of human attraction to infantile features [49], extrinsic qualities of human personalities [55] and a susceptibility to fashion trends [40,58] fueled in part by media exposure [62,63]. The prominence of the internet as a favored source of information consulted in the acquisition process has also been highlighted in this section [65,66]. Further investigation of the motives and pre-purchase behaviors underpinning choices with regard to dog type would benefit from greater attention to the interaction of cultural and demographic factors.

## 4. Choosing Which Individual Dog to Acquire

### 4.1. Similarities in Appearance and Personality between Owner and Dog

There is evidence to suggest that the decision to acquire a specific dog might be influenced by the degree of similarity between the physical appearance and personality of individual dogs, and that of the owner [69,70]. In studies exploring this trend, participants were asked to attempt to match photographs of dogs with their owners. The photographs used in Payne and Jaffe’s [70] study were cropped to show only the faces of humans and dogs, while Roy and Christenfeld [69] used photographs of owners from the waist up, and those of dogs showing the whole dog, facing forward. In both studies, a resemblance was reported between purebred dogs and their owners. However, when the resemblance between non-purebreds and their owners was investigated, no resemblance was found. These results support the hypothesis that the ability to match owner and dog is a consequence of selection rather than convergence, as no association was observed between the ability to pair an owner with his or her pet and the duration of time they had cohabited. While it might be hypothesized that this finding could reflect a lesser concern for the importance of the dog’s appearance among owners who acquire their dog through a shelter or rescue (rather than from a breeder of purebred dogs), the findings of Weiss et al.’s aforementioned study [43] highlight the dog’s appearance as the most cited single reason for adopters choosing their dog from the shelter environment. However, the final appearance of a purebred puppy is much more predictable than that of a non-purebred. Overall, the results of the photo matching studies suggest that people select dogs that, at some level, look like them, both in terms of their facial features and their broader physical features (e.g., size, hair, and attractiveness), and that when a purebred is acquired, people tend to get what they want in this regard. Payne and Jaffe [70] conclude that the psychological mechanisms guiding pet choice resemble those guiding human mate choice: namely, assortative mating. The level at which the resemblance between owner and dog exists is not explained by these studies. Whether the resemblance exists at the level of physical attributes, such as size, or at a stylistic level, such as a friendly appearance, could be profitably further explored in future studies. In addition, a more nuanced understanding of the importance that adopters of non-purebreds report to confer on appearance when choosing their dog would be worthy of future study, to investigate why owner–dog resemblance does not appear to exist in these circumstances.

The association between pet choice and owner identity is considered to run deeper than at a merely superficial level however. A person’s choice of pet may be a representation of the owner’s self [71] and owners might project their self-identity onto their pets [72]. In 1985, Veevers [73] suggested that people keep pets as a medium of expression for the personality and preferences of the owner. Recent research conducted by Tesfom and Birch [39] indicates that this assertion might still apply. In their study they reported an association between how dog owners define themselves and their dog breed choice; overall, those who perceived their personal behavior and race as important in defining their sense of self rated the dog’s breed as an important feature in their acquisition choice.

### 4.2. Canine Behavior and Temperament

Various studies conducted within the shelter environment have linked the behavior or personality of dogs to the likelihood of their selection and length of time to their adoption [4,74]. In a survey conducted among members of the public in Belfast, Northern Ireland, “temperament” was regarded as the most important factor when selecting a dog from a rescue center [74]. In this study, participants were also asked to select their preferred photograph of a dog from a choice of two that differed in one specific way. The results suggest that dogs at the front of their kennels, as well as those that are labelled as quiet as opposed to barking, may garner more adopter interest and preference than those at the back. The findings also suggest that people are more attracted to dogs that have a toy in their kennel, even though the dog was not shown to be interacting with the toy. However, the indirect methodology of this study and the use of participants who were not necessarily actively engaged in the acquisition process, presents limitations to the application of these findings. Rather than measuring how dog behavior impacts on human preferences, in this case it appears to be people’s perceptions and assumptions about a dog’s behavior that are under investigation. 

More recently, and using more direct methods of measurement, a study found that the kinds of behaviors displayed by dogs (*n* = 289) in shelter kennels influenced their length of stay in the shelter, [4]. To collect data, an observer video-recorded dogs in their kennels for one minute on five days per week over eight months. After controlling for factors that may influence the duration of stay, such as small size, long coat, or preferred breed type (ratter, herder or lap), it was found that leaning or rubbing on the kennel wall, facing away from the front of the kennel, and standing all increased a dog’s length of stay. Dogs that exhibited back and forth movement in their kennel added around 15–25 days to their length of stay. While several behaviors were found to predict a longer length of stay, the study found that no behaviors predicted a shorter length of stay across the population studied. Thus, they hypothesized that adopters were more sensitive to undesirable than to desirable behaviors, especially those that indicated low sociability. 

The character of a dog’s interactions with prospective adopters has also been found to be an important factor to influence decisions. Protopopova and Wynne [75] observed the behavior of shelter dogs (*n* = 151) during out-of-kennel interactions with potential adopters (*n* = 250), identifying two behaviors that they suggest impact adoption decisions: the dog’s willingness to respond to the adopter’s play initiation, and the dog’s propensity to lie close to the adopter. For instance, dogs that were adopted spent, on average, twice as much time lying close to the adopter than dogs that were not adopted. In another study, researchers asked adopters, at the time of adoption, what behaviors their pet displayed during their first meeting [43]. Adopters stated that their dogs approached and greeted, licked, jumped on them, and wagged their tails during their first meeting. Most of the adopters (78%) also reported that the dog’s personality or temperament was an important reason in their choice to adopt the individual dog. The researchers thus concluded that the behaviors reported by the adopters might have influenced their choices. However, as the study did not collect comparative data regarding the experiences of people who visited but chose not to adopt, it is not possible to determine a causal relationship between the dog’s behavior and their likelihood of adoption.

While most research exploring the importance of canine behavior in attracting people to dogs has been conducted within the shelter environment, this aspect has also been identified as important among dog owners more generally. In an Australian survey, both dog-owning and non dog-owning members of the public were asked to rate the importance of various physical and behavioral characteristics as they pertained to their image of the ideal dog [44]. The behavioral traits rated most often as being extremely important to owners’ descriptions of their ideal dog included a desire for the dog to be safe with children, to be housetrained, to come when called, to enjoy being petted and to display affection to their owners. These findings are consistent with those from a similar version of this study conducted in Italy [45]. 

The studies discussed in this section appear to indicate that physical appearance attracts some people to choose individual dogs [43]. For owners of purebred dogs, there seems to be a particular motivation to acquire dogs that resemble, at some level, the physical characteristics of the owner [69,70]. However, while physical appearance is consistently identified as an important determinant among owners, canine temperament and behavior—including both the owner’s perceived temperament of the breed, and the behaviors displayed by the dog in direct engagement with prospective owners—might also be significant in acquisition decisions [4,43,44,45,74]. It is worth noting, however, that the studies measuring participants’ ratings of their preferred dog attributes provide evidence based on preference or intention, rather than behavior. Although intention and preference may predict later behavior, this distinction must nevertheless be acknowledged and limits the application of this evidence beyond the context of intent. Of the studies discussed that investigated the behaviors of dogs available for acquisition, the populations of dogs studied were limited to those within the shelter environment. Further research might explore whether similar findings occur in dogs within breeding establishments. In addition, to develop an understanding of whether a causal relationship exists between dog behavior and acquisition, future studies would benefit from paying attention to the experiences of those who visit rescue organisations/breeders and interact with available dogs but do not choose to adopt/acquire a dog. Finally, future study in this area should explore whether, and how, dog behavior influences male and female prospective owners differently.

## 5. Choosing Where to Acquire a Dog

According to recent statistics gathered by PDSA (2018) [65], most people in the UK obtain their puppies from breeders (28%), rescue or rehoming centers (22%), or private sellers (20%). The RSPCA (2018) [76] also reported that 22% of UK dog owners acquire dogs from rescue centers. However, they estimate that a greater proportion of owners obtain dogs from breeders (44%). In the US, according to a national survey of pet owners conducted by the American Pet Products Association (APPA) [1], the most common sources for the acquisition of pet dogs were breeders (25%), friends or relatives (25%), animal shelters/humane societies (22%) or rescue groups (12%). 

Drawing comparisons between the different estimates is challenging as the surveys use different menus of defined response choices. For instance, in contrast to the PDSA report, the category “private seller” was not listed in the RSPCA or APPA reports. In addition, owners may not always be clear on how to accurately define their mode of acquisition. Beyond the Kennel Club affiliated breeders, it may not be apparent to purchasers whether the individual offering a puppy for sale is a professional (i.e., licensed) breeder, or an individual engaged in illicit breeding and trading activities. Unscrupulous breeders and dealers have been reported to go to considerable lengths to present themselves as reputable; sometimes renting houses in an effort to reassure potential buyers that the puppy comes from that home, when in reality it has been transported in from a less desirable source [77].

Similarly to decision-making around choice of dog type or breed, evidence suggests that people’s decisions about which sources to consider acquiring their dog from are likely associated with a complex variety of underlying beliefs, values and socioeconomic and demographic factors. 

### 5.1. Human Values and Beliefs

In a 2016 study by Bir, Widmar and Croney [78], the most commonly reported reason for obtaining a dog in a particular manner was that “it was the right thing to do” (47%). This finding must be interpreted with some caution, as while this study’s sample comprised respondents who had acquired dogs through various means, including both purchase and adoption, the most popular mode of acquisition among this sample was adoption from a shelter/rescue organisation (39%). 

While certain ethical values might guide people towards acquiring a dog through the mode of adoption, other perceptions about the perceived behavior of dogs from a shelter environment might discourage people from considering adoption. For instance, one-third of surveyed current and potential dog owners in Australia believe that dogs from shelters often have behavior problems [79]. This study also found that ambivalent perceptions about shelter dogs are likely to affect people’s decision about whether to consider rehoming rather than purchasing. For example, many respondents chose the “neither agree nor disagree” option for statements such as: “Dogs in shelters were often mistreated in their previous home” and “Adult shelter dogs often have behavior problems”.

### 5.2. A Desire for a Specific Type or Breed

In Bir et al.’s study [78], the owner’s preference for a specific breed or type of dog (e.g., purebred, mixed breed or designer breed) (33%) was the second most common reason given, after ethical motivations, for owners acquiring their dog in a particular manner. One way in which this might impact the source from which people acquire dogs is through a belief that rescue centers are unable to offer their desired breed. Indeed, in the findings of both Maddalena, Zeidman and Campbell’s [80] and Bir et al.’s [78] studies, common reasons people gave for *not* wanting to acquire a dog from a rescue center included their desire for a “purebred” dog and being uncertain that a shelter could provide the type of dog they desired. In King et al.’s [44] study regarding the ideal companion dog, 41% of respondents indicated that their ideal dog would be purebred, while the type of dog was reported as unimportant to 43% of the sample. Regarding the desire for a purebred dog, research conducted in a rescue environment in the UK has indicated a positive bias towards purebred dogs, finding that purebred dogs are typically adopted sooner than crossbreeds and mixed breeds [81,82]. 

### 5.3. Socioeconomic and Demographic Factors

Several American studies have observed gender differences with regard to source preference. Bir et al. [56] found that women were more inclined to favor adoption/rescue as a mode of obtaining a dog when compared to men, with a significantly higher percentage of women (40%) than men (32%) agreeing that the *only* responsible way to acquire a dog is through a shelter/rescue. Similarly, Reese, Skidmore, Dyar and Rosebrook [83] found women to be significantly more likely than men to acquire dogs for the purpose of rescuing them. In a study exploring the predominance of women as dog rescue workers in Michigan, USA, Queen [84] considered political affiliation as an independent variable. The study’s findings indicate that, among other factors, the typical dog rescue worker is more likely to be a Democrat. Although Bir et al. [78] collected data on respondents’ political affiliation in their study of dog acquisition, their analysis does not report on this variable. Future research could therefore benefit from including political affiliation as a factor to explore.

Bir et al. [56] also identified age as an important demographic characteristic related to perceptions of the various sources for dog acquisition. In particular, the beliefs of older respondents (aged 55–88 years) were frequently significantly different to the other two, younger, age categories (18–34 and 35–54 years). For instance, those in the older category were more inclined than younger people to believe that the acquisition of purebred dogs is acceptable and appeared more supportive of the notion that people should have choices as to where and which dogs they obtain (provided those dogs were not imported from overseas). The researchers speculated that some of these differences might be associated with differences in experience, in addition to a potentially heightened sensitivity of younger respondents to some dog welfare issues (e.g., high dog euthanasia rates in the US). This finding is in line with the findings of another recent study, by Woodhead, Feng, Howell, Ruby and Bennett [85], that explored how beliefs about dog-breeding and acquisition varied among a sample of Australian adults (*n* = 986). Based on survey-based data pertaining to various factors—including degree of endorsement of statements regarding dog breeding practices, knowledge of dog breeding, and history of dog ownership—the researchers identified three groups of respondents: “improve breeding”, “breeding as is” and “exclusive adoption”. The members within each cluster shared similar beliefs about dog breeding and acquisition, with several demographics found to vary between the groups. One of the factors that was found to distinguish members of the “exclusive adoption” group was their age, as they were significantly younger than the members of the other two groups. 

The effect of income level on source preference has been variably reported. In Reese et al. [83], people with an income of less than $20,000 were significantly more likely than those in other income groups to have acquired their dog from a family member or another person. Meanwhile, respondents with very high incomes (>$150,000) were significantly more likely than those in other income groups to have obtained their dog from a breeder. However, in Woodhead et al.’s [85] study of perceptions of dog breeding practices, breeding dog welfare and dog acquisition, annual household income was not found to be statistically significant with regards to the three groups of respondents identified in their analysis. 

Regarding owner demographics, one factor that has been investigated in relation to perceptions and behavior around sources of dogs is education level. Most recently, the results of Woodhead et al. [85] did not find an association between level of education and attitudes towards practices of dog breeding, welfare and acquisition. However, other studies do suggest an association between acquisition behavior and education level. In Bir et al. [56], respondents with at least a college degree were more likely to adopt from a shelter or rescue center than those without college degrees. However, caution is required in interpreting Bir et al.’s results as the study reports solely on respondent’s degree of agreement with statements related to dog acquisition and welfare aspects, rather than their actions. This distinction is important, especially given that research on this topic has found a disconnect between what dog owners say they would do and what behaviors they have demonstrated in practice. For instance, Garrison and Weiss [86] found that although 60% of respondents who had acquired their dogs in the last year reported that they had considered adopting from an animal shelter, only 39% of those people had obtained their dog from such a source. In Reese et al.’s study [83], which investigated actual methods of acquiring dogs, people with college degrees were more likely to adopt from a shelter or rescue than those without a college education, in keeping with Bir et al.’s findings [56]. 

### 5.4. Barriers to Adoption as a Mode of Acquisition

Greater insight as to why people might report that they would consider shelters but do not end up acquiring dogs through this means would be valuable. A recent study conducted amongst American pet owners indicated several barriers to adoption as a chosen mode of acquisition [87]. Reflecting on the stage of conducting pre-purchase research, owners expressed difficulty in finding a pet they wanted, and, furthermore, finding a pet geographically close to them. Further barriers were identified when prospective adopters contacted the shelter or rescue center, including: no/slow response from the organisation; details of pets up for adoption not being kept up to date; and, the requisite for potential adopters to submit applications prior to being given further details on the animal. When completing an application, additional barriers were reported: the length of the application process and the requirements imposed, frequent refusal of applications due to potential adopter’s working hours or lack of garden, and requirements for adopters to wait whilst their application is processed and home-checks made. Thus, at any stage of the acquisition process, a potential adopter might become frustrated and choose to leave the adoption process in favor of purchasing an animal from a breeder, private seller or pet store.

When choosing where to get a dog from, the evidence suggests that this decision is firstly impacted by whether a prospective owner has strong feelings with regard to acquiring a specific breed or type of dog [44,78,80]. Among those for whom this factor is important, beliefs about whether a particular source is likely able to supply them with their preferred type or breed appear to take precedence. For those who do not express a desire for a specific breed or type, the ethical impulse to adopt a dog in need is found to be of considerable importance in shaping decisions to adopt (i.e., acquire a rescue) rather than buy (i.e., from a breeder) [78]. The studies discussed also indicate that a preference towards adopting/rescuing dogs as a mode of dog acquisition is especially pronounced in women [56,83], in younger populations [56] and in populations with a lower annual income (less than $20,000) [82]. One limitation of some of the studies explored here is a reliance on data regarding respondent’s intentions rather than actual behavior. While intentions might often predict later behavior, with regards to dog ownership, this may vary depending on factors such as how soon into the future people are intending to acquire a dog. Future studies exploring dog acquisition could benefit from a pre-post design that collects data regarding the intentions of people who are planning on acquiring a dog within the next year, and then follows up with finding out which dogs were acquired and how they were sourced. 

## 6. Conclusions

The studies considered in this review highlight multiple factors that likely impact decision-making surrounding dog acquisition. Human-related factors include prior dog ownership experience, age, gender, ethnicity, income, education and household structure. Various demographic factors, such as gender, age and income, have also been identified as potential predictors of preference or behavior with regards to choosing where to acquire a dog from. In addition, social influences, such as trends in breed popularity, appear to influence decisions regarding which breed of dog to acquire. The findings explored in this review also indicate the importance of several dog-related factors that commonly impact prospective owner’s decision-making. Most notably, the dog’s physical appearance, temperament and behavior have all been found to be of significance with regards to owner preferences and acquisition behavior. While breed health is a determinant of the decision to acquire a dog for some owners, overall the dog’s appearance seems to be more important than the dog’s health. This finding has been observed among certain populations of owners, for instance owners of particular breeds such as the French Bulldog.

While the studies provide support for the prominent role that certain owner demographics and dog characteristics appear to have with regards to influencing the motivations and behaviors of prospective dog owners, this review has highlighted the need for further research to elucidate the interaction of these factors with greater precision. The studies considered in this review had a range of important limitations which must be acknowledged. Most of the data reported were retrospective and many of the findings based on correlations. Thus, the causality of any associations identified remains largely unestablished. Further well-designed studies are, therefore, required in order to build knowledge that will help to shape tools and strategies for the effective education of prospective owners, to promote responsible acquisition practices, and to help those working in the dog rescue environment to improve their ability to attract suitable prospective owners and promote canine welfare. Additionally, this discussion has emphasized the need for future study to understand what environmental factors, beyond knowledge alone, drive human behavior, with the aim to develop strategies to encourage people to exhibit responsible behaviors in the dog acquisition process.

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
