# Peer review of "Acquiring a Pet Dog: A Review of Factors Affecting the Decision-Making of Prospective Dog Owners"

_animals, 2019, doi:10.3390/ani9040124_

Round 1
Reviewer 1 Report
I think the title could do with having the words “A review” in it, to be clear at a glance that this is a review article for anyone searching for it.
Line 34: remove the ‘as to’ from before ‘whether’
Line 36: what does ‘type’ mean here?
End of Intro: It’s also worth mentioning how some owners may make impulse purchases based on opportunity, and as such will not experience many of these stages fully. For example, for someone who’s friend/family member or mate down the pub offers them a puppy from their own litter, they may accept on impulse, perhaps even without consulting other household members (something my Dad did!), in which case no decisions have been made about breed/type, or where to acquire it have been made, and there is unfortunately no scope for intervention in such cases.
Socioeconomic status: This factor must interact with the others (such as house ownership) in some or many ways. I think one of the issues people have probably encountered is that the relationship is unlikely to be linear i.e. there will be a point at which having a higher income means less dog ownership, because people will be working/out of the house too much to be able to care for a dog. These studies also fail to consider (correct me if I’m wrong) the frequency of dog ownership amongst homeless populations.
Line 243: “supports Lorenz’s claim” reads oddly as ‘Lorenz’ hasn't been mentioned previously.
Line 453: remove the second ‘study’
Line 543: I think the first word here is meant to be ‘acquisition’ not adoption.
I found the Conclusions section to be highly unsatisfactory, sorry. Whilst the author has done a good job of presenting the key findings of lots of different studies on this subject, the conclusions don’t synthesise the review for the reader: What are the key things the author has learned from all of those studies? What are the take-home messages you want the reader to remember from your review? At the moment the conclusions mainly just point to limitations and a need for further work, which aren't true conclusions and could lead a reader to assume that nothing of value can be learnt from reviewing the existing literature. I would greatly like to see this section re-written and expanded upon to synthesise the main messages/findings from this rather long review: it currently lacks the authors findings from doing this review.
Author Response
22 March 2019
Re: Acquiring a Pet Dog: Factors Affecting the Decision-Making of Prospective Owners (ID: animals-439366)
Dear Reviewer,
Thank you very much for taking the time to engage with my paper. I greatly appreciate the comments and suggestions provided and have revised the manuscript accordingly. Please see below, in blue, my point-by-point responses to the comments. All line numbers refer to the revised manuscript file with tracked changes.
Sincerely,
Katrina
Dr Katrina Holland
Dogs Trust
17 Wakley Street
London EC1V 7RQ
Response to Reviewer 1 comments:
I think the title could do with having the words “A review” in it, to be clear at a glance that this is a review article for anyone searching for it.
Thank you for this suggestion. I have revised the title to highlight that it is a review article. The title now reads: ‘Acquiring a Pet Dog: A Review of Factors Affecting the Decision-Making of Prospective Dog Owners’.
Line 34: remove the ‘as to’ from before ‘whether’
Thank you for pointing out this grammatical error. ‘as to’ has been removed. (Line 34)
Line 36: what does ‘type’ mean here?
‘Type’ is included to refer to breed category. For instance, Protopopova et al. (2014) grouped together breeds into seven ‘types’: Fighting, Herding, Hound, Lap, Ratters, Sporting, Working. This clarification has been added where ‘type’ is first used. (Lines 36-37)
End of Intro: It’s also worth mentioning how some owners may make impulse purchases based on opportunity, and as such will not experience many of these stages fully. For example, for someone who’s friend/family member or mate down the pub offers them a puppy from their own litter, they may accept on impulse, perhaps even without consulting other household members (something my Dad did!), in which case no decisions have been made about breed/type, or where to acquire it have been made, and there is unfortunately no scope for intervention in such cases.
Thank you for this suggestion and raising this important point. I have added mention of this to the introduction, where the topic of dog acquisition is outlined to the reader. (Lines 39-43)
Socioeconomic status: This factor must interact with the others (such as house ownership) in some or many ways. I think one of the issues people have probably encountered is that the relationship is unlikely to be linear i.e. there will be a point at which having a higher income means less dog ownership, because people will be working/out of the house too much to be able to care for a dog. These studies also fail to consider (correct me if I’m wrong) the frequency of dog ownership amongst homeless populations.
Thank you for this comment. At the end of section 2.1.3. I suggest that proxies of SES, as predictors of dog acquisition behaviour, are likely to interact with other factors. (Lines 159-162)
It is true that the studies included do not consider rates of dog ownership amongst homeless populations and this is an important area that requires further research. I have included this observation in the discussion on accommodation type. (Lines 111-114)
Line 243: “supports Lorenz’s claim” reads oddly as ‘Lorenz’ hasn't been mentioned previously.
Thank you for pointing this out. Although Lorenz’s work was cited in the previous sentence, his name was not previously included in this sentence so it was not explicit. I have revised to include his name in the citation, making the first reference explicit. (Line 305)
Line 453: remove the second ‘study’
Thank you for pointing out this typo. Second ‘study’ has been removed. (Line 615)
Line 543: I think the first word here is meant to be ‘acquisition’ not adoption.
Thank you for pointing out this typo. ‘Adoption’ changed to ‘acquisition’. (Line 712)
I found the Conclusions section to be highly unsatisfactory, sorry. Whilst the author has done a good job of presenting the key findings of lots of different studies on this subject, the conclusions don’t synthesise the review for the reader: What are the key things the author has learned from all of those studies? What are the take-home messages you want the reader to remember from your review? At the moment the conclusions mainly just point to limitations and a need for further work, which aren't true conclusions and could lead a reader to assume that nothing of value can be learnt from reviewing the existing literature. I would greatly like to see this section re-written and expanded upon to synthesise the main messages/findings from this rather long review: it currently lacks the authors findings from doing this review.
Thank you for this comment. I acknowledge that the conclusion may have been unsatisfactory and have rewritten this section to synthesise the findings discussed in this paper. I have revised the manuscript to include an explicit summary of the studies discussed in each section (Choosing to acquire a dog; Choosing which type of dog to acquire; Choosing which individual dog to acquire; Choosing where to acquire a dog from), highlighting key findings, limitations and areas for future research. (Lines 199-209; 451-464; 569-588; 726-742).
The conclusion has been re-written to bring together key findings and limitations from the studies explored throughout. (Lines 745-779)

Reviewer 2 Report
Please see attached

Author Response
22 March 2019
Re: Acquiring a Pet Dog: Factors Affecting the Decision-Making of Prospective Owners (ID: animals-439366)
Dear Reviewer,
Thank you very much for taking the time to engage with my paper. I greatly appreciate the comments and suggestions provided and have revised the manuscript accordingly. Please see below, in blue, my point-by-point responses to the comments. All line numbers refer to the revised manuscript file with tracked changes.
Sincerely,
Katrina
Dr Katrina Holland
Dogs Trust
17 Wakley Street
London EC1V 7RQ
Response to Reviewer 2 comments:
L60 – Below you begin discussing research into the demographic and socioeconomic breakdown of people who acquire dogs. It would be useful for you to highlight this discussion here as it doesn’t fit clearly into motivations or behaviours, but is a very important area to discuss.
Thank you for this suggestion. I have revised the manuscript, highlighting this area of the discussion within the introduction. (Lines 65-66)
L102 – The use of the paper’s year is awkward, consider removing. “Inadequate power of the study [15] to detect a difference in a small and relatively homogenous study area” works fine.
Thank you for this suggestion. I have removed the date as suggested. (Line 107)
L109 – Many readers may not understand the difficulty in measuring SES so it is worth putting in a line or two here about that, to explain why a proxy is used.
Thank you for this suggestion. I agree that the difficulty in measuring SES might not be well understood by all readers and have included several lines at the beginning of this paragraph clarifying my definition of SES, explaining why SES is difficult to measure and why proxies are used. (Lines 118-123)
L125 – Are these in order of perceived SES? I am wondering if management/self employed fits in terms of this finding.
Thank you for this comment. The categorizations used to indicate SES (blue collar; white collar; management/self-employed), as included in the study referenced (Almqvist et al. 2003), were given in order of perceived SES (i.e. blue collar referred to lowest socioeconomic index). However, the study does not make explicit the method used to classify occupations into these categories. To avoid any confusion, I have amended the discussion of this study, removing reference to these categorizations. Instead, the manuscript now simply states the study’s finding that a low socioeconomic index was associated with a greater likelihood of dog ownership. (Lines 147-150)
L127 – As this is the most commonly understood indication of SES, I would consider putting this ahead of occupation and education.
Thank you for this suggestion. I agree that the discussion would be better ordered as you suggest and have reordered this section to present income/affluence as the first measure discussed. (Line 124)
L133 – Summary should begin as a new paragraph
Thank you for this comment. I have amended so the summary of these studies begins as a new paragraph. (Line 157)
L151 – What a sentence! To enhance readability, I would revise the way this section begins.
Thank you for this suggestion. I agree that this sentence was difficult to read and have revised the beginning of this section to make it more comprehensible. (Lines 176-182)
L160 – I think this section would definitely benefit from a summary of what the research is telling us, the limitations in general, and what we need to do next to further our understanding of this area. I have skipped down to your conclusion and you have a sentence on this, but I think your paper could benefit from an explicit summary here.
Thank you for this suggestion. I have added an explicit summary of the research discussed in this section. Summaries for each section have been added throughout the paper. (Lines 199-209)
L196 – There were some other findings in this study that are worth highlighting here also, such as the overall popularity of shorter hair and dog size.
Thank you for this suggestion. This study did report several other interesting findings not included in the original manuscript. I have revised the manuscript to include mention of these other findings and others from a similar study conducted in Italy (Diverio et al. 2016). (Lines 241-244)
Reference
Diverio, S.; Boccini, B.; Menchetti, L.; Bennett, P.C. The Italian perception of the ideal companion dog. Journal of Veterinary Behavior 2016 12: 27-35.
L236 – Anything else? Above you’ve alluded to multiple behaviours and set this paragraph up really nicely to discuss this study’s findings in a bit more detail.
Thank you for this comment. Packer et al.’s study raises various interesting findings with regards to the acquisition behaviour of owners of brachycephalic dogs in comparison to owners of non-brachycephalic dogs and I have revised the manuscript to include discussion of these. (Lines 291-300)
L255 – You may not have room, but I think this is a great spot to consider WHY they’ve become more popular. The breed has been around for a while and human preferences for infantile features is not a new phenomenon. You could simply say you discuss breed popularity further down, but give an example of why frenchies may have become so popular in such a short space of time – perhaps celebrity purchases.
Thank you for this suggestion. I agree that considering why they have become more popular is relevant for the discussion and I have added a few sentences to this paragraph considering why their popularity might have increased over recent times. (Lines 324-332)
L269 – Did they find any evidence to support this? Perhaps highlight if not that more research needs to be conducted to investigate this theory.
Thank you for this suggestion. The study found some evidence for the suggestion raised. However, given the exploratory nature of the research methods, more research is certainly required. I have added more detail regarding the study’s findings and pointed to its limitations and the need for further research regarding this theory to consider cultural and demographic differences. (Lines 344-353)
L304 – Also worth mentioning is individuals capitalising on breed popularity and demand by producing higher volumes of puppies, possibly leading to decreased welfare of breeding dogs and their puppies.
Thank you for this suggestion. This is an important issue and I have added in a few sentences on this to the revised manuscript. (Lines 389-400)
L368 – It is super interesting that this finding was not observed in those owners of mixed breed dogs. Perhaps this is because those more likely to purchase a dog from a shelter or rescue (i.e. not a breeder) find the dogs appearance less important. Is there any evidence to support this? Perhaps loosely Woodhead et al. (2018), which you discuss further down.
Thank you for this comment. I agree that this finding is very interesting. However, I have not found any evidence to support the theory that those who purchase from a shelter/rescue find the dog’s appearance less important than those who purchase from a breeder (either in Woodhead et al. (2018) or elsewhere). Weiss et al. (2012) investigated the reasons why adopters chose their dog (from an animal shelter in the USA), finding that the dog’s appearance was cited most often as the single most important reason people adopted their dog (27%). When asked in a multiple-choice format what factors were important for adopters in their decision to choose particular dogs, 75% reported physical appearance. This was followed by personality/temperament (16%) and behavior with people (11%).
I have extended the discussion of the finding regarding owner/dog resemblance in the revised manuscript and highlighted what evidence is missing from this topic. (Lines 478-500)
L373 – There has been some research done regarding a person’s political affiliation and where they would purchase their dog from – I believe Bir et al. (https://www.ncbi.nlm.nih.gov/pmc/articles/PMC5575571/) covered it. It would be worth discussing here if you have the room to do so.
Thank you for this comment. In Bir et al.’s 2017 paper that you mention, they do not report on political affiliation. I think the research you may be thinking of is a preliminary report of their findings (Bir et al. 2016), available here. In this, the study authors include respondents’ political affiliation within their table of summary statistics. However, political affiliation is not a variable reported on in the main analysis, so its role is unknown.
Beyond dog acquisition specifically, in a study of dog rescue workers in Michigan, Queen (2014) considered political affiliation as an independent variable that might help explain the predominant role of women in the world of dog rescue. Fewer men reported being Democrats than would be expected and Queen’s findings suggest that, among other factors, the typical rescuer is more likely to be a Democrat.
I have included reference to these studies, and the potential for future research to explore the impact of political affiliation, in section 5. (Lines 655-660)
References
Bir, C.; Widmar, N.J.O.; Croney, C.C. Public perceptions of dog acquisition: Sources, rationales and expenditures. Center for Animal Welfare Science at Purdue University 2016. Available online: https://vet.purdue.edu/CAWS/files/documents/20160602-public-perceptions-of-dog-acquisition.pdf (accessed on December 2018).
Queen, R. The overwhelming predominance of women in the world of dog rescue: the state of Michigan as a representative case study enhanced by relevant interview data from rescuers elsewhere. In American Dog Rescue and the Discourse of Compassion; Markovits, A.; Crosby, K. (Eds.); University of Michigan Press: Michigan, 2014; 64-79.
L446 – Also the buyers don’t know exactly how to define their acquisition – particularly around what is characterised as a breeder beyond the kennel club registered ones.
Thank you for this comment. It is an important point to note which I have added to the revised manuscript. (Lines 602-609)
L530 – This is a very important point to make and needs to be taken into consideration with majority of the research in this area. Although intent is commonly predictive of later behaviour, I think this may differ in a population of people thinking about their future dog and those looking to acquire a dog very soon or in the near future. An interesting study would be pre-post design talking to people who are looking to acquire a dog within the next year, and then follow up with what dog they got/where from etc.
Thank you for this comment. I agree that this kind of study would help to overcome the limitations of other studies based on data regarding the intentions of prospective owners, and I have added this suggestion in the summary of section 5. (Lines 737-743)
L554 – Similar to the comment left regarding L160, I think that each section – ‘choosing to acquire a dog’, ‘Choosing which type of dog to acquire’, ‘Choosing which individual dog to acquire’, and ‘5. Choosing where to acquire a dog from’ would all benefit from a short summary of the findings, limitations and future directions. Then your conclusion could bring all such summaries together to provide a comprehensive overview and illustrate the next steps.
Thank you for this suggestion. Summaries for each section have been added throughout the revised manuscript. (Lines 199-209; 451-464; 569-588; 726-742).
The conclusion has been re-written to bring together key findings and limitations from the studies explored throughout. (Lines 745-779)

Round 2
Reviewer 1 Report
All of my comments have been addressed and the manuscript has been significantly improved by the addition of true conclusions, thank you.